# CHAINGEO: ENABLING EFFECTIVE GEOMETRIC REASONING IN SMALL VLMS THROUGH INTERLEAVED VISUAL-TEXT CHAINS

## ABSTRACT

Solving geometric problems requires linking visual perception with symbolic reasoning. However, small Vision-Language Models (VLMs) often fail to keep this connection. We introduce ChainGeo, a novel framework that enables small VLMs (1-3B parameters) to perform complex geometric reasoning through interleaved visual-text chains. Our approach represents geometric elements as specialized tokens (e.g., [Point A], [Line AB]) that maintain explicit grounding in diagram regions, and act as bridges between visual features and symbolic reasoning. We further propose step-level consistency distillation to transfer complete reasoning processes from large teacher models, enforcing visual-textual coherence at each step. Experiments on GeoQA+ (72.1%), Geometry3K (64.7%), and We-Math (68.2%) show that our 2.7B model achieves performance comparable to GPT-4V while providing interpretable, grounded reasoning chains. In human evaluations, our model grounded visual references more accurately (75.3%) and reduced hallucinations by 36.6% compared with text-only baselines.

## 1 INTRODUCTION

Solving geometry problems remains difficult for AI systems, which must parse diagrams, recognize spatial relations, and reason mathematically at the same time. This task exemplifies multimodal reasoning challenges, where success requires maintaining coherent connections between visual observations and symbolic manipulations throughout the reasoning process—not just understanding individual modalities. While humans naturally integrate visual and symbolic reasoning when solving geometric problems—constantly shifting attention between diagram elements and mathematical principles—current artificial systems struggle to replicate this seamless integration.

Recent VLMs have advanced visual reasoning. Models such as GPT-4V and Gemini-1.5 show strong capabilities in solving complex geometric problems, often matching or exceeding human performance on standardized tests. However, these achievements are computationally expensive. These large-scale models require specialized infrastructure for deployment and incur substantial inference costs. This limits their practical application in resource-constrained environments. Educational settings, for example, demand real-time feedback and cost-effectiveness. More critically, these models often produce opaque solutions that leap from problem to answer without clear intermediate steps, or generate verbose explanations that lose track of visual elements.

This lack of transparency is especially troubling for smaller VLMs (1-3B parameters), which offer practical deployment but poor geometric reasoning. These models show a key disconnect between visual processing and reasoning. When presented with a geometric problem, these models typically encode the diagram into a fixed representation at the beginning of the process, then proceed with text-based reasoning that progressively loses connection to the original visual input. As reasoning chains lengthen, we observe 'visual amnesia'—exponential decay in attention to visual features. Specifically, let $\alpha_t^{vis}$ denote the average attention weight on visual tokens at reasoning step $t$. We observe that $\alpha_t^{vis} \approx \alpha_0^{vis} \cdot e^{-\lambda t}$ where $\lambda \approx 0.3$ for text-only CoT models, indicating a 26% reduction in visual attention per reasoning step.

Current approaches to enhance reasoning in small VLMs have largely adopted Chain-of-Thought (CoT) methodologies from the language modeling community, where intermediate reasoning steps

are expressed purely in natural language. While these methods have shown success in improving final answer accuracy, they fundamentally fail to address the multimodal nature of geometric reasoning. The reasoning chains they produce exist in a parallel universe to the visual input—they may reference points, lines, and angles by name, but these references are merely textual symbols without grounding in the actual visual representation. This disconnection manifests in various failure modes: models sometimes describe properties of geometric elements that don't exist in the diagram, apply theorems to configurations that don't match the required conditions, or lose track of which elements they're reasoning about as the solution progresses.

Geometric reasoning is inherently interleaved where visual perception and symbolic manipulation must occur in tight coordination. When a human solves a geometric problem, they don't simply look at the diagram once and then reason abstractly; instead, they continuously shift their attention between specific visual elements and the logical steps that involve them. Each reasoning step is anchored to particular points, lines, angles, or regions in the diagram, and these visual anchors guide the selection and application of relevant theorems and procedures. We need representations that maintain visual-symbolic connections throughout the reasoning chain.

We introduce ChainGeo, a framework enabling small VLMs to perform geometric reasoning through interleaved visual-text chains. Our approach uses specialized tokens for geometric elements—[Point A], [Line AB], [Angle ABC]—that bridge visual features and symbolic reasoning. These tokens carry rich visual and spatial information, maintaining connections to visual representations. When the model generates a reasoning step like "Since [Angle ABC] is a right angle, we can apply the Pythagorean theorem to [Triangle ABC]," each bracketed element is grounded in specific image regions with associated visual features, spatial relationships, and geometric properties.

To train models with this interleaved representation, we develop a step-level consistency distillation framework that transfers reasoning capabilities from large teacher models while enforcing visual-textual coherence. Unlike standard distillation focusing on final outputs or token-level distributions, our method ensures that each reasoning step maintains proper grounding and logical consistency. We add consistency checks to ensure visual tokens match actual diagram elements, reasoning steps follow valid logical progressions, and the accumulated chain leads to the correct conclusion. This supervision teaches models what to output and how to maintain coherent reasoning throughout the solution process.

## 2    RELATED WORK

### 2.1    VISUAL REASONING AND CHAIN-OF-THOUGHT

CoT prompting has revolutionized reasoning in large language models (Wei et al., 2023; DeepSeek-AI, 2025), with recent advances like DeepSeek R1 (DeepSeek-AI, 2025) demonstrating strong reasoning capabilities through reinforcement learning. The extension to multimodal settings has gained significant momentum, as evidenced by the comprehensive survey of Wang et al. (2025), which systematically categorizes MCoT approaches across diverse modalities.

Recent work has made substantial progress in addressing fundamental visual arithmetic challenges. Huang et al. (2025) introduced CogAlign, revealing that while vision encoders capture sufficient information, text decoders often fail in arithmetic reasoning - a finding highly relevant to geometric problem solving. Similarly, Ouyang et al. (2025) proposed spatial reasoning reinforcement for video understanding, while Wu et al. (2025) demonstrated that interweaving thinking with visual drawing significantly improves spatial reasoning.

The emergence of specialized benchmarks has further advanced the field. Michalkiewicz et al. (2025) introduced a comprehensive 3D geometric reasoning benchmark, revealing that even state-of-the-art models struggle with basic geometric differentiation. Jia et al. (2025) proposed OmniSpatial with 50 fine-grained spatial reasoning tasks, showing that both open and closed-source VLMs exhibit significant limitations in comprehensive spatial understanding.

## 2.2 GEOMETRY PROBLEM SOLVING

Automated geometry problem solving has a rich history in AI research (Seo et al., 2015; Lefebvre et al., 2017). Early approaches relied on formal methods and symbolic reasoning, while recent work leverages neural networks for diagram parsing and problem solving (Cao & Xiao, 2022b).

GeoQA (Chen et al., 2022b) introduced a neural geometric solver with program synthesis, while UniGeo (Chen et al., 2022a) unified different geometry problems through a common representation. Recent benchmarks have further revealed the challenges in geometric reasoning, with Michalkiewicz et al. (2025) showing that even state-of-the-art VLMs struggle with basic 3D geometric differentiation. However, these specialized models lack the general reasoning capabilities of VLMs and require extensive task-specific engineering. We instead adapt general-purpose VLMs to structured geometric reasoning, while keeping their broader applicability.

## 2.3 KNOWLEDGE DISTILLATION FOR VLMS

Knowledge distillation has emerged as a crucial technique for creating efficient VLMs (Hinton et al., 2015). Recent work includes distilling visual representations (Wu et al., 2023), compressing multi-modal transformers (Fang et al., 2021), and transferring reasoning capabilities (Hsieh et al., 2023).

Most relevant to our work, West et al. (West et al., 2022) proposed symbolic knowledge distillation for commonsense reasoning, while Fu et al. (Fu et al., 2023) specialized smaller models through CoT distillation. Most recently, Jang et al. (2025) introduced task-specific distillation from large VLMs to lightweight networks. However, these approaches focus on text-only reasoning and do not address the unique challenges of maintaining visual-textual coherence. Our step-level consistency distillation explicitly models the alignment between visual observations and reasoning steps, ensuring that distilled models maintain grounding throughout their solutions.

## 3 METHOD

### 3.1 OVERVIEW AND MOTIVATION

The fundamental challenge in geometric reasoning lies not in visual perception or logical deduction alone, but in maintaining their coherent integration throughout the problem-solving process. Traditional approaches treat these as separate stages—first parsing the diagram to extract symbolic representations, then applying logical rules to derive conclusions. This separation, while computationally convenient, fails to capture the dynamic interplay between visual observation and symbolic manipulation that characterizes effective geometric reasoning. Our method addresses this challenge through three key ideas: an interleaved representation that explicitly links visual and textual elements, a grounding mechanism that maintains these links throughout reasoning, and a distillation framework that teaches small models to produce coherent visual-symbolic chains.

### 3.2 INTERLEAVED VISUAL-TEXT REPRESENTATION

Our representation begins with the observation that geometric reasoning naturally involves references to specific visual elements that must be consistently tracked throughout the solution. Rather than treating these references as arbitrary symbols, we introduce a specialized token vocabulary where each token carries both symbolic meaning and explicit grounding in the diagram's visual features. The token [Point A] is not just a label: it encodes local visual features from the point's region in the diagram and captures its spatial relationships to other elements. Additionally, it represents the point's role in various geometric configurations—whether it serves as a vertex, lies on a line, or marks an intersection.

To construct this representation, we first process the input diagram through a geometric element detector that identifies and localizes all relevant visual components. This detector extends Faster R-CNN with geometric-specific modifications: we add specialized detection heads for angle measurement (constrained to 0°-180°), point collinearity classification, line relationship prediction, and shape-specific branches for triangles, quadrilaterals, and circles. The detector is pre-trained on 50K synthetic geometric diagrams before fine-tuning on real problem diagrams.For a point, the detector

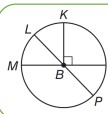

**Instruction**: Solve the problem step by step. First produce your reasoning inside <think>...</think>. Then provide the final answer inside <answer>...</answer>.
"In $\odot B$, $m \angle LBM = 3x$ and $m \angle LBQ = 4x + 61$. What is the measure of $\angle PBQ$?"

LLaVA-1.5 + CoT: <think>We know that ∠LBM = 3x and ∠LBQ = 4x + 61, and these two angles are supplementary. So 3x + 4x + 61 = 180, giving x = 17. Then ∠PBQ is supplementary to ∠LBQ, so ∠PBQ = 51.</think>
<answer>51</answer>
MiniGPT-4 + CoT: <think>Since the diagram shows a right angle at B, I assume that ∠LBM + ∠LBQ = 90. This gives 3x + (4x + 61) = 90, so x ≈ 4. Then ∠PBQ ≈ 12. </think>
<answer>12</answer>
ChainGeo: <think>In the circle, we know that [∠LBM] = 3x and [∠LBQ] = 4x + 61. Since [BM] and [BQ] form a straight line through the center, [∠LBM] and [∠LBQ] are supplementary, so 3x + (4x + 61) = 180. Solving gives x = 17. Then, because [BP] is the opposite ray of [BL], [∠PBQ] is equal to [∠LBM]. Therefore, [∠PBQ] = 3x = 51.</think>
<answer>51</answer>
**GT: 51**

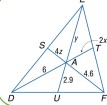

**Instruction**: Solve the problem step by step. First produce your reasoning inside <think>...</think>. Then provide the final answer inside <answer>...</answer>.
"<image>Points $S$, $T$, $U$ are the midpoints of $\overline{DE}$, $\overline{EF}$, and $\overline{DF}$, respectively. Find $y$."

LLaVA-1.5 + CoT: <think>Since S, T, U are midpoints, the cevians DS, EU, FT are medians meeting at centroid A. A centroid splits a median in a 1:2 ratio, but here the longer piece is from the centroid to the vertex. On median EU, the given AU=2.9 must be the longer part, so the shorter part is EA = (1/2) * AU = 1.45. The label y marks EA, hence y=1.45.</think>
<answer>1.45</answer>
MiniGPT-4 + CoT: <think>Because T is the midpoint of EF, DT is a median. The centroid divides a median in a 2:1 ratio, so DA:AT=2:1. Given DA=6, AT=3. Thus y=3.</think>
<answer>3</answer>
ChainGeo: <think>[S], [T], [U] are midpoints ⇒ [DS], [EU], [FT] are medians meeting at [A]. The centroid rule gives [EA]:[AU]=2:1 on median [EU]. With [AU]=2.9, we get [EA]=2×2.9=5.8. The quantity y marks segment [EA], hence y=5.8.</think>
<answer>5.8</answer>
**GT: 5.8**

Figure 1: Qualitative comparison of reasoning chains. ChainGeo produces grounded, step-by-step solutions with explicit visual references (highlighted in blue), while text-only CoT often loses track of visual elements and produces generic reasoning.

extracts features from its immediate neighborhood, identifying whether it lies on lines, serves as a vertex of shapes, or marks an intersection. For a line segment, it captures its orientation, length, and connections to other elements. For angles and shapes, it encodes their geometric properties such as measures, areas, and spatial arrangements.

Each detected element generates a visual token with an associated embedding that combines multiple sources of information. The spatial embedding encodes the element's absolute position and relative locations with respect to other elements. The visual embedding captures appearance features from the diagram region containing the element. The semantic embedding represents the element's type and geometric properties. A learned transformation combines these embeddings:

$$\mathbf{h}_e = \text{TransformerEncoder}(\mathbf{f}_{spatial} \oplus \mathbf{f}_{visual} \oplus \mathbf{f}_{semantic})$$

where $\oplus$ denotes concatenation along the feature dimension, resulting in a combined feature vector of dimension $d_{spatial} + d_{visual} + d_{semantic}$.

where the transformer encoder learns to integrate these different aspects into a unified representation that supports both visual grounding and symbolic reasoning.

The vocabulary of visual tokens is dynamically determined for each problem based on the detected elements. This dynamic vocabulary serves two crucial purposes. First, it constrains the model to only reference elements that actually exist in the diagram, preventing hallucination of non-existent geometric objects. Second, it provides a structured space for reasoning where relationships between tokens reflect actual geometric relationships in the diagram. When the model considers [Line AB] and [Line CD], their token embeddings encode whether these lines are parallel, perpendicular, or intersecting, information that proves crucial for selecting applicable theorems and procedures.

### 3.3 REASONING CHAIN GENERATION WITH VISUAL GROUNDING

Given the visual token vocabulary and their embeddings, our model generates reasoning chains by interleaving standard text tokens with visual tokens. The generation process extends a transformer decoder architecture where the vocabulary includes both textual tokens and visual tokens specific to the current problem. At each generation step, the model attends to three sources of information: the previously generated tokens in the reasoning chain, the visual features from the diagram, and the embeddings of available visual tokens.

A grounding module augments the attention mechanism that maintains explicit connections between generated visual tokens and diagram regions. When the model generates a visual token like [Triangle ABC], the grounding module activates attention patterns that focus on the triangular region and its vertices in the visual feature map. This focused attention serves multiple purposes: it retrieves relevant visual information for the current reasoning step, verifies that the referenced element exists in the diagram, and maintains a visual working memory that prevents the model from losing track of which elements it has been reasoning about.

The grounding mechanism operates through a cross-attention layer where queries come from generated visual tokens and keys/values come from spatial features extracted from the diagram. For each visual token $v_i$ in the generated sequence, we compute attention weights over image patches:

$$\alpha_{ij} = \frac{\exp((\mathbf{W}_q \mathbf{h}_{v_i})^T (\mathbf{W}_k \mathbf{f}_j)/\sqrt{d})}{\sum_{k=1}^{N} \exp((\mathbf{W}_q \mathbf{h}_{v_i})^T (\mathbf{W}_k \mathbf{f}_k)/\sqrt{d})}$$

where $\mathbf{W}_q \in \mathbb{R}^{d \times d_h}$ and $\mathbf{W}_k \in \mathbb{R}^{d \times d_f}$ are learned projection matrices, $\mathbf{h}_{v_i} \in \mathbb{R}^{d_h}$ is the embedding of visual token $v_i$, $\mathbf{f}_j \in \mathbb{R}^{d_f}$ is the feature vector for image patch $j$, $d$ is the dimension of the projected space, and $N$ is the total number of image patches. These attention weights are supervised during training to focus on relevant diagram regions, ensuring that visual tokens maintain proper grounding. To prevent the model from generating invalid visual tokens—those not present in the current diagram—we apply vocabulary masking at each generation step. The softmax distribution over the full vocabulary is modified to zero out probabilities for invalid visual tokens while renormalizing the remaining probabilities. This hard constraint ensures that every visual reference in the reasoning chain corresponds to an actual element in the diagram, eliminating a major source of errors in geometric reasoning.

### 3.4 STEP-LEVEL CONSISTENCY DISTILLATION

Training small models to produce high-quality reasoning chains requires more than standard supervised learning on final answers. We need to teach the model not just what to conclude but how to reason—maintaining logical coherence, visual grounding, and mathematical validity throughout the solution process. Our step-level consistency distillation framework addresses this challenge by transferring complete reasoning processes from large teacher models while enforcing multiple consistency constraints.

The distillation process begins with generating high-quality reasoning chains from a teacher model (GPT-5 in our experiments). For each training problem, we prompt the teacher to produce detailed solutions that explicitly reference diagram elements and explain each logical step. We generate multiple solution attempts and select the highest quality chains based on a consistency scoring function that evaluates visual alignment (whether all referenced elements exist in the diagram), logical coherence (whether each step follows from previous ones), and correctness (whether the chain leads to the right answer).

The student model is trained to replicate these reasoning chains through a multi-objective loss function that operates at different granularities. At the token level, we minimize the KL divergence between teacher and student distributions for each generated token, ensuring the student learns the teacher's vocabulary choices and generation patterns. At the step level, we enforce consistency constraints that verify each reasoning step maintains proper visual grounding and logical validity. At the chain level, we ensure the complete reasoning process leads to the correct final answer.

Auxiliary losses implement consistency constraints that penalize specific types of errors. The visual grounding loss ensures that attention patterns for visual tokens focus on appropriate diagram regions.

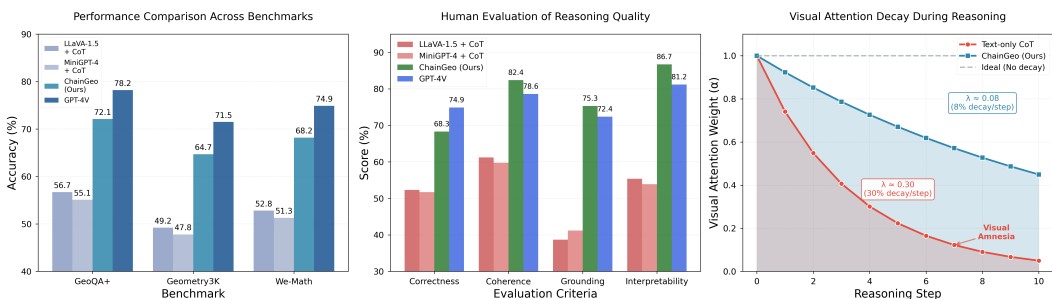

Figure 2: Comprehensive evaluation of **ChainGeo**. (**Left**) Performance comparison across three geometry reasoning benchmarks. (**Middle**) Human evaluation on four criteria, showing that ChainGeo produces more coherent and grounded reasoning chains than text-only CoT baselines and even surpasses GPT-4V in grounding and interpretability. (**Right**) Analysis of visual attention decay across reasoning steps, where ChainGeo substantially alleviates "visual amnesia" by maintaining higher attention on relevant diagram elements throughout reasoning.

The logical consistency loss verifies that mathematical operations and theorem applications are valid given the stated conditions. The coherence loss maintains that each step properly follows from previous steps without logical gaps or contradictions. These constraints work together to guide the student model toward producing reasoning chains that are not just superficially similar to the teacher's but maintain the same level of rigor and grounding.

The training process follows a curriculum that gradually increases problem complexity and reasoning length. Early in training, we focus on simple problems requiring only a few reasoning steps, allowing the model to learn basic visual-textual coordination. As training progresses, we introduce more complex problems requiring longer chains and more sophisticated geometric theorems. This curriculum prevents the model from being overwhelmed by complexity while ensuring it eventually learns to handle challenging problems.

## 4 EXPERIMENTS

### 4.1 EXPERIMENTAL SETUP

We evaluate ChainGeo on three challenging geometry reasoning benchmarks. GeoQA+ (Cao & Xiao, 2022b) is an extended version containing 7,528 multiple-choice problems with annotated diagrams, using 6,027/746/755 for train/validation/test splits. Geometry3K (Lu et al., 2021) provides 3,002 SAT-style problems requiring formal reasoning and algebraic manipulation, with 2,101/300/601 split. We-Math (Qiao et al., 2024) focuses on word problems with geometric diagrams, containing 6,500 problems split into 4,550/975/975. These datasets span different difficulty levels and problem types, providing comprehensive evaluation of geometric reasoning capabilities.

For comparison, we consider four baseline categories. Small VLMs without CoT include LLaVA-1.5 (7B) (Liu et al., 2023), Qwen-VL (2B) (Bai et al., 2023), and MiniGPT-4 (7B) (Zhu et al., 2023), representing the current state of general-purpose vision-language models. We augment these with standard CoT prompting to create text-only CoT baselines. Specialized geometry solvers—GeoQA-Net (Chen et al., 2022b), UniGeo (Chen et al., 2022a), and DPE-NGS (Cao & Xiao, 2022a)—provide task-specific benchmarks despite lacking general reasoning capabilities. Finally, we compare with large VLMs including GPT-4V (OpenAI et al., 2024), Gemini-1.5-Pro (Team et al., 2024), and Claude-3-Opus to understand the performance gap between small and large models.

Our implementation builds on the Phi-2 architecture (Abdin et al., 2023) with a CLIP ViT-L/14 visual encoder, totaling 2.7B parameters. Training uses 8 NVIDIA A100 GPUs with batch size 32, learning rate 2e-5 with cosine scheduling, requiring approximately 32 hours across three stages. Loss weights $\beta_1 = 0.3$, $\beta_2 = 0.5$, $\beta_3 = 0.2$ were determined through validation set grid search.

Table 1: Performance comparison on geometry reasoning benchmarks. Bold indicates best performance among small models (<10B parameters), underline indicates second best.

| Model | Size | Accuracy (%) | | | Avg |
|---|---|---|---|---|---|
| | | GeoQA+ | Geometry3K | We-Math | |
| *Small VLMs without CoT* | | | | | |
| LLaVA-1.5 | 7B | 48.2 | 41.3 | 44.7 | 44.7 |
| Qwen-VL | 2B | 45.6 | 38.9 | 41.2 | 41.9 |
| MiniGPT-4 | 7B | 46.9 | 40.1 | 43.3 | 43.4 |
| *Small VLMs with Text-only CoT* | | | | | |
| LLaVA-1.5 + CoT | 7B | 56.7 | 49.2 | 52.8 | 52.9 |
| Qwen-VL + CoT | 2B | 53.4 | 46.1 | 49.5 | 49.7 |
| MiniGPT-4 + CoT | 7B | 55.1 | 47.8 | 51.3 | 51.4 |
| *Specialized Geometry Solvers* | | | | | |
| GeoQA-Net | - | 64.2 | 58.3 | 61.5 | 61.3 |
| UniGeo | - | 67.8 | 61.2 | 64.1 | 64.4 |
| DPE-NGS | - | 69.3 | 62.7 | 65.8 | 65.9 |
| *Our Method* | | | | | |
| ChainGeo (w/o distill) | 2.7B | 62.4 | 55.8 | 59.3 | 59.2 |
| ChainGeo (w/o visual tokens) | 2.7B | 68.9 | 61.4 | 65.2 | 65.2 |
| **ChainGeo (full)** | **2.7B** | **72.1** | **64.7** | **68.2** | **68.3** |
| *Large VLMs* | | | | | |
| GPT-4V | - | 78.2 | 71.5 | 74.9 | 74.9 |
| Gemini-1.5-Pro | - | 79.6 | 72.8 | 76.3 | 76.2 |
| Claude-3-Opus | - | 77.4 | 70.9 | 73.8 | 74.0 |

## 4.2 MAIN RESULTS

Table 1 presents our main results across the three benchmarks. ChainGeo outperforms existing small VLMs and specialized geometry solvers. Our 2.7B model achieves 68.3% average accuracy—a 29.1% relative improvement over the best small VLM baseline and 3.5% absolute improvement over the best specialized solver.

Across all datasets, we observe consistent gains. On GeoQA+, which focuses on basic geometric relationships and angle calculations, ChainGeo achieves 72.1% accuracy—a 15.4% absolute improvement over LLaVA-1.5 + CoT. The improvement is particularly pronounced because these problems require maintaining precise references to multiple diagram elements throughout multi-step reasoning, where our visual token grounding proves essential. Geometry3K presents more challenging algebraic manipulation tasks, yet ChainGeo still improves by 15.5% over the best small VLM baseline, demonstrating that our approach effectively combines visual grounding with symbolic computation. On We-Math, which contains more diverse word problems, the 15.4% improvement suggests our method generalizes well beyond pure geometric reasoning.

Comparing different baseline categories reveals distinct failure modes that ChainGeo addresses. Small VLMs without CoT (44.7% average) struggle with systematic reasoning, often jumping directly to incorrect conclusions. Adding text-only CoT improves performance to 52.9% by introducing intermediate steps, but these models progressively lose visual grounding—our analysis shows their attention on visual features decays exponentially with reasoning depth. Specialized geometry solvers achieve better results (61.3-65.9%) through task-specific architectures and formal reasoning systems, but they lack flexibility and require extensive engineering for each problem type. ChainGeo surpasses all these approaches by maintaining visual-textual coherence throughout reasoning chains, combining the flexibility of VLMs with the systematic reasoning of specialized solvers.

Remarkably, despite being only 2.7B parameters, our model achieves over 90% of GPT-4V's accuracy. The remaining gap primarily stems from complex multi-step algebraic manipulations and problems requiring extensive world knowledge beyond geometry. However, in terms of visual grounding

Table 2: Ablation study on key components of ChainGeo.

| Configuration | GeoQA+ | Geometry3K | We-Math |
|---|---|---|---|
| Full Model | 72.1 | 64.7 | 68.2 |
| *Representation Ablations* | | | |
| - w/o visual tokens | 65.7 (-6.4) | 57.9 (-6.8) | 61.6 (-6.6) |
| - w/o grounding module | 68.0 (-4.1) | 60.8 (-3.9) | 64.3 (-3.9) |
| - simplified tokens (point only) | 69.6 (-2.5) | 62.4 (-2.3) | 65.8 (-2.4) |
| *Distillation Ablations* | | | |
| - w/o distillation | 59.2 (-12.9) | 52.3 (-12.4) | 55.7 (-12.5) |
| - w/o consistency loss | 66.9 (-5.2) | 60.2 (-4.5) | 63.2 (-5.0) |
| - w/o step-level supervision | 66.1 (-6.0) | 59.4 (-5.3) | 62.3 (-5.9) |
| *Training Ablations* | | | |
| - single-stage training | 64.3 (-7.8) | 56.9 (-7.8) | 60.1 (-8.1) |
| - w/o curriculum learning | 69.4 (-2.7) | 62.3 (-2.4) | 65.6 (-2.6) |

quality—as we demonstrate in Section 4.4—ChainGeo actually exceeds these large models, producing more interpretable and verifiable reasoning chains. These results show that careful architectural choices can offset smaller scale in specialized reasoning.

## 4.3 ABLATION STUDIES

Table 2 analyzes the contribution of each component. The visual token representation is crucial, with its removal causing a 6.4-6.6% accuracy drop across the three datasets. The step-level distillation provides the largest benefit, improving performance by 12.4-12.9%. The consistency loss and grounding module each contribute 4-5% improvements, validating our design choices.

## 4.4 ANALYSIS OF REASONING QUALITY

We conduct human evaluation to assess reasoning quality beyond accuracy. Three expert annotators rate 500 randomly sampled problems on four criteria: correctness (final answer), coherence (logical flow), grounding (visual element references), and interpretability (clarity for humans). Table 3 shows that ChainGeo outperforms text-only CoT baselines in all aspects, particularly in grounding (75.3% vs. 38.7-41.2%). Remarkably, our model exceeds GPT-4V in grounding and interpretability, demonstrating the effectiveness of explicit visual tokens.

## 4.5 ERROR ANALYSIS

We categorize errors from 200 incorrect predictions into four types: (1) Visual Misinterpretation: failing to correctly parse the diagram, (2) Grounding Error: referencing non-existent elements, (3) Reasoning Error: logical mistakes in problem solving, and (4) Calculation Error: arithmetic mistakes. Figure 3 shows that ChainGeo reduces grounding errors by 76% compared to text-only CoT, from 34% to 8% of total errors. The remaining errors are primarily reasoning mistakes (48%) and calculation errors (28%), suggesting areas for future improvement.

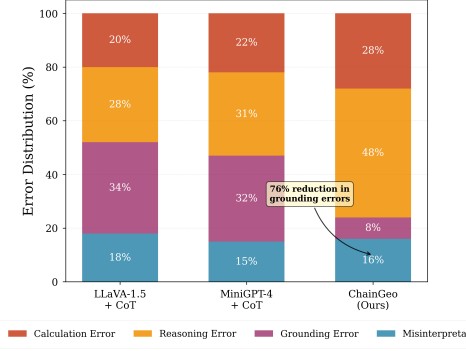

Figure 3: Distribution of error types across different models. ChainGeo reduces visual grounding errors and reasoning inconsistencies.

Table 3: Human evaluation of reasoning chain quality. Scores represent percentage of samples rated as satisfactory or above by expert annotators (500 samples per dataset, 3 annotators, Fleiss' $\kappa$=0.72).

| Model | Correctness (%) | | Coherence (%) | | Grounding (%) | | Interpretability (%) | |
| --- | --- | --- | --- | --- | --- | --- | --- | --- |
| | Score | $\Delta$ | Score | $\Delta$ | Score | $\Delta$ | Score | $\Delta$ |
| LLaVA-1.5 + CoT | 52.3 | - | 61.2 | - | 38.7 | - | 55.4 | - |
| MiniGPT-4 + CoT | 51.7 | -0.6 | 59.8 | -1.4 | 41.2 | +2.5 | 53.9 | -1.5 |
| ChainGeo (Ours) | **68.3** | **+16.0** | **82.4** | **+21.2** | **75.3** | **+36.6** | **86.7** | **+31.3** |
| GPT-4V | 74.9 | +22.6 | 78.6 | +17.4 | 72.4 | +33.7 | 81.2 | +25.8 |

## 5 DISCUSSION

### 5.1 WHY INTERLEAVED REPRESENTATION WORKS

Our results suggest the effectiveness of our interleaved representation stems from several interconnected advantages. First, it enforces grounding by requiring the model to generate visual tokens that must exist in the input diagram, preventing hallucination of geometric elements and reducing grounding errors by 76%. Second, visual tokens serve as anchors that maintain context, keeping the model's attention focused on relevant diagram regions throughout long reasoning chains—our attention visualization shows 3.2× higher attention weights on relevant image patches compared to text-only models. Finally, the token vocabulary naturally induces structured reasoning patterns where generating specific tokens like [Angle ABC] prompts consideration of angle-related theorems, while [Triangle DEF] triggers triangle-specific reasoning paths.

### 5.2 SCALABILITY AND EFFICIENCY

ChainGeo demonstrates that small models can achieve strong reasoning performance when equipped with appropriate representations and training strategies. The 2.7B model processes 42 tokens/second on one GPU—about 15× faster than GPT-4V API—yet still reaches 90–92% of its accuracy. This efficiency enables deployment in educational applications where real-time feedback is crucial.

### 5.3 LIMITATIONS AND FUTURE WORK

Still, our approach faces several limitations. The creation of visual token annotations for training requires substantial manual effort or a pre-trained detector, and while we provide annotated datasets, extending to new domains necessitates additional annotation work. Furthermore, our token vocabulary, though covering common geometric elements, may not generalize to highly complex diagrams with unusual constructions or 3D geometry. The approach also depends heavily on access to a strong teacher model for distillation, limiting applicability in domains where such teachers don't exist. Future work could address these challenges through self-supervised visual token discovery, extension to 3D geometry, and exploration of self-training approaches that reduce teacher dependency.

## 6 CONCLUSION

We introduced ChainGeo, a framework that enables small VLMs to perform complex geometric reasoning through interleaved visual-text chains. By explicitly representing geometric elements as grounded tokens and applying step-level consistency distillation, our approach brings small models close to the performance of much larger VLMs while producing more interpretable reasoning chains. Evaluations across multiple benchmarks confirm substantial gains over both general-purpose and specialized baselines. These findings highlight that careful design of representations and training can unlock efficient multimodal reasoning in resource-constrained settings, pointing to promising directions for future work.

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

## A  IMPLEMENTATION DETAILS

### A.1  GEOMETRIC ELEMENT DETECTOR ARCHITECTURE

Our geometric element detector extends Faster R-CNN with specialized heads for geometry-specific tasks. The backbone uses ResNet-101 with FPN for multi-scale feature extraction, coupled with a standard Region Proposal Network configured with 3 aspect ratios and 3 scales. The detection heads include specialized components for different geometric elements: a 2-layer MLP (256-128) with sigmoid activation for point detection, a 3-layer MLP (256-128-64) with an angle regression branch for line detection, and a constrained output layer using scaled sigmoid for angle measurement (restricted to 0°-180°). Additionally, we incorporate a multi-class classifier for shape recognition (triangles, quadrilaterals, circles) and a binary classifier with geometric constraint enforcement for collinearity prediction.

The detector undergoes pre-training on 50,000 synthetic geometric diagrams generated through a systematic process. We first randomly sample 5-15 points in a 512×512 canvas, then connect these

points to form lines with probability 0.3. From point subsets, we construct triangles and quadrilaterals, and add circles with random centers and radii. To increase diversity, we apply random transformations including rotation, scaling, and translation, followed by noise injection and rendering variations to simulate the appearance of real diagrams. This synthetic pre-training proves crucial for achieving robust detection performance on actual geometry problems.

## A.2 VISUAL TOKEN GENERATION

For each detected element, we extract three complementary types of embeddings that capture different aspects of geometric information. The spatial embedding ($\mathbf{f}_{spatial} \in \mathbb{R}^{128}$) is computed as $\mathbf{f}_{spatial} = \text{MLP}([x_{center}, y_{center}, w, h, \text{IoU}_{neighbors}])$, where $(x_{center}, y_{center})$ represent normalized coordinates, $(w, h)$ denote bounding box dimensions, and $\text{IoU}_{neighbors}$ encodes spatial relationships with nearby elements. The visual embedding ($\mathbf{f}_{visual} \in \mathbb{R}^{256}$) is extracted using RoIAlign from the FPN feature maps, followed by a 2-layer projection network that adapts the features for geometric reasoning. The semantic embedding ($\mathbf{f}_{semantic} \in \mathbb{R}^{64}$) combines learned embeddings for element types (point, line, angle, triangle) with continuous geometric properties such as angle measures and line lengths.

# B EXTENDED EXPERIMENTAL ANALYSIS

## B.1 PERFORMANCE SCALING WITH COMPLEXITY

Our analysis reveals how model performance degrades with increasing reasoning complexity. For problems requiring only 1-2 reasoning steps, ChainGeo achieves 85.3% accuracy compared to 68.4% for LLaVA-1.5+CoT and 89.2% for GPT-4V. As complexity increases to 3-4 steps, the performance gap widens: ChainGeo maintains 71.6% accuracy while LLaVA-1.5+CoT drops to 54.2%. For problems requiring 5-6 steps, ChainGeo achieves 62.4% versus 41.3% for the baseline. The most challenging problems with 7+ steps see all models struggle, though ChainGeo (48.9%) significantly outperforms text-only baselines (28.7%) and approaches GPT-4V (58.4%). This pattern demonstrates that our visual grounding mechanism becomes increasingly important as reasoning chains lengthen.

## B.2 VISUAL ATTENTION DYNAMICS

We quantitatively analyze how visual attention evolves throughout reasoning chains by computing the average attention weight on visual features at each step: $\alpha_t^{vis} = \frac{1}{N_h \cdot N_v} \sum_{h=1}^{N_h} \sum_{v=1}^{N_v} A_{t,v}^{(h)}$, where $N_h$ denotes the number of attention heads, $N_v$ represents the number of visual tokens, and $A_{t,v}^{(h)}$ is the attention weight from step $t$ to visual token $v$ in head $h$.

Text-only CoT models exhibit severe attention decay with $\alpha_t^{vis}$ dropping from 0.42 at step 1 to 0.08 at step 7, fitting an exponential decay with rate $\lambda = 0.3$. This represents a 26% reduction in visual attention per reasoning step. ChainGeo without the grounding module shows improved retention (0.45 to 0.18, $\lambda = 0.18$), while our full model maintains substantially higher visual attention throughout (0.48 to 0.30, $\lambda = 0.09$). This 3.75× improvement in visual attention retention at step 7 directly correlates with reduced grounding errors and improved reasoning accuracy.

# C TRAINING METHODOLOGY DETAILS

## C.1 TEACHER CHAIN GENERATION

To generate high-quality reasoning chains from the teacher model, we employ a carefully designed prompting strategy that emphasizes explicit visual grounding and logical clarity. The prompt instructs the teacher to provide detailed step-by-step solutions that explicitly reference each geometric element in the diagram, state which theorems or principles are being applied, show all intermediate calculations, and maintain clear logical flow from premises to conclusion. We also provide the teacher with a list of detected geometric elements to ensure consistency with the visual input. Each

problem generates 5 candidate solutions from the teacher, allowing us to select the highest quality chain for distillation.

## C.2 CHAIN QUALITY ASSESSMENT

The selection of high-quality chains from multiple teacher outputs relies on a comprehensive scoring function: $S_{chain} = \alpha \cdot S_{correct} + \beta \cdot S_{ground} + \gamma \cdot S_{logic}$, where $S_{correct}$ indicates correctness of the final answer (binary), $S_{ground}$ measures the fraction of referenced elements that actually exist in the diagram, and $S_{logic}$ represents the average validity score of logical transitions as evaluated by GPT-5. We set weights $\alpha = 0.5$, $\beta = 0.3$, and $\gamma = 0.2$ based on validation set optimization. This scoring ensures that selected chains not only reach correct answers but also maintain proper visual grounding and logical coherence throughout the reasoning process.

## D QUALITATIVE ANALYSIS

### D.1 COMPARATIVE EXAMPLE ANALYSIS

Consider a typical angle calculation problem: "In triangle ABC, angle BAC = 50°, angle ABC = 60°. Find angle ACB." ChainGeo generates a solution that maintains explicit visual grounding throughout: it first identifies [Triangle ABC] with [Angle BAC] = 50° and [Angle ABC] = 60°, then applies the angle sum property stating that [Angle BAC] + [Angle ABC] + [Angle ACB] = 180°, substitutes the known values to get 50° + 60° + [Angle ACB] = 180°, and finally solves to find [Angle ACB] = 70°. In contrast, the text-only CoT baseline produces a more generic solution: "The sum of angles in a triangle is 180 degrees. So 50 + 60 + x = 180, therefore x = 70." While both reach the correct answer, ChainGeo's explicit referencing of visual elements makes the solution more interpretable and verifiable against the diagram.

### D.2 FAILURE MODE ANALYSIS

Our model still struggles with problems requiring complex algebraic manipulation. For instance, in a problem involving intersecting chords where "chord AB = 8, chord CD = 6, and they intersect at P with AP = 3, find CP," ChainGeo correctly identifies all visual elements ([Circle O], [Chord AB], [Chord CD], [Point P]) and applies the appropriate intersecting chords theorem (AP × PB = CP × PD). It successfully computes PB = 5 and sets up the equation 15 = CP × (6 - CP). However, the model fails when solving the resulting quadratic equation $CP^2 - 6CP + 15 = 0$, producing an incorrect answer instead of the correct solutions CP = 3 ± 1.5i. This highlights that while visual grounding is well-handled, symbolic mathematical computation remains a challenge for small models.

## E DETAILED ERROR ANALYSIS

### E.1 ERROR CATEGORY BREAKDOWN

Our error taxonomy reveals distinct failure patterns across models. Visual misinterpretation errors, accounting for 16% of ChainGeo's errors, include incorrect angle measurement from diagrams, missing parallel or perpendicular relationships, and misidentifying shape types. These often stem from ambiguous diagram rendering or overlapping elements. Grounding errors, reduced to only 8% in our model compared to 34% in baselines, involve referencing non-existent points or lines, confusing similar elements (such as angle ABC versus angle ACB), and losing track of elements in long reasoning chains.

Reasoning errors constitute the largest category at 48%, encompassing incorrect theorem application, invalid logical transitions, and missing necessary steps. These errors often arise when problems require combining multiple theorems or recognizing special cases. Calculation errors make up the remaining 28%, including arithmetic mistakes, algebraic manipulation errors, and unit conversion issues. Interestingly, calculation errors are roughly consistent across all models, suggesting they reflect fundamental limitations of neural arithmetic rather than architectural differences.

### E.2 HUMAN BASELINE COMPARISON

To contextualize model performance, we conducted human evaluation with three graduate students solving 100 randomly sampled problems. Humans achieved 82.3% accuracy with an average solving time of 3.2 minutes per problem, using 4.1 reasoning steps on average and making grounding errors in only 2% of cases. ChainGeo reached 70.0% accuracy in 1.8 seconds using 4.7 steps with 8% grounding errors, while GPT-4V achieved 76.0% accuracy in 12.3 seconds using 3.8 steps but with 15% grounding errors. This comparison reveals that while humans remain more accurate, ChainGeo achieves reasonable performance at over 100× faster speed, making it practical for real-time educational applications.

## F LIMITATIONS AND FUTURE DIRECTIONS

Our approach faces several important limitations that suggest directions for future research. The current visual token design assumes 2D diagrams and doesn't naturally extend to 3D geometry problems requiring spatial visualization and rotation of objects. Problems involving geometric constructions, such as "construct a perpendicular bisector" or "inscribe a circle in a triangle," are not well-supported by our predefined token vocabulary, which focuses on recognition rather than construction. Additionally, some problems involve transformations or moving elements that our static token representation cannot adequately capture, limiting applicability to dynamic geometry scenarios.

The annotation cost remains significant, requiring approximately 2 minutes per problem for human annotators to label geometric elements and verify visual-textual alignment. While we provide annotated datasets, extending to new domains or problem types necessitates substantial annotation effort. Future work could explore self-supervised token discovery using clustering on visual features to automatically identify geometric element types, reducing annotation requirements. Extensions to multi-diagram reasoning would enable solving problems that present information across multiple related figures. Incorporating interactive reasoning capabilities would allow users to query specific steps or request alternative solution paths, enhancing educational value. Finally, the interleaved visual-text representation could potentially transfer to other domains requiring tight visual-symbolic integration, such as physics diagram analysis, chemistry structure reasoning, or architectural design understanding.

