# OpenReview forum: "ChainGeo: Enabling Effective Geometric Reasoning in Small VLMs through Interleaved Visual-Text Chains"
_ICLR.cc/2026/Conference — Submitted to ICLR 2026_

### Official Review · Reviewer_a2DU · 2025-10-29

**Soundness:** 1
**Presentation:** 2
**Contribution:** 1
**Rating:** 0
**Confidence:** 4

**Summary:**

This paper propose to solve geometric problems through symbolic reasoning in smale VLM, thus build ChainGeo framework that enables complex geometric reasoning. The central idea is the specialized geometric tokens to represent symbolic elements, and consistency distillation from large VLM. Extensive experiments were conducted on three datasets with improved performance.

**Strengths:**

The technical contributions are very limited and the experimental settings are seriously flawed.

**Weaknesses:**

Presentation：

--The writing is poor and difficult to follow.

Method：

--Technical contribution is poor;

--Limited novelty of representing geometric symbols as special tokens and distilling similarities.

Experiment:

--The compared large VLMs (e.g. GPT 4V, Gemini1.5) are all outdated, making the results unreliable.

--The compared specialized geometric solvers are all from before 2022.

--The authors only compare a limited number of early works, making it difficult to assess its contribution.

**Questions:**

--If the large VLMs possesses a strong geometric reasoning ability, what are the motivations and application scenarios for distilling down to a smaller model?

--Why not conduct experiments and comparisons based on methods from recent years?

---

### Official Review · Reviewer_USH7 · 2025-10-30

**Soundness:** 3
**Presentation:** 3
**Contribution:** 3
**Rating:** 6
**Confidence:** 3

**Summary:**

This paper introduces ChainGeo, a framework designed to enhance complex geometric reasoning in small Vision-Language Models (1~3B parameters) by addressing the disconnect between visual perception and symbolic reasoning. It uses interleaved visual-text chains—specialized tokens grounded in diagram regions—to bridge visual features and logical reasoning, paired with step-level consistency distillation that transfers structured reasoning processes from large teacher models. ChainGeo enables small vision-language models to achieve geometric reasoning performance on par with large models such as GPT-4V, while being significantly more computationally efficient with inference speeds 15 times faster and producing reasoning chains that are more interpretable and verifiable. The approach is rigorously validated across multiple benchmarks, outperforming both general-purpose small VLMs and specialized geometry solvers.

**Strengths:**

1. The paper innovatively introduces interleaved visual-text reasoning chains into small-scale vision-language models (1~3B parameters), explicitly anchoring symbolic reasoning to visual elements throughout the inference process. Unlike traditional text-only chain-of-thought (CoT) approaches, this representation systematically addresses— for the first time—the problem of "visual amnesia" in geometric reasoning with small models.
2. The approach converts geometric elements (points, lines, angles, shapes) into explicit visual tokens and binds them to corresponding image regions, providing an interpretable interface for vision-symbol integration. This token-level grounding can be viewed as a novel multimodal intermediate representation, offering methodological innovation.
3. ChainGeo provides a systematic implementation pathway that enables vision-language models to explicitly align visual elements with symbolic reasoning steps, laying a foundation for future research in areas such as 3D geometry and physical scene understanding.

**Weaknesses:**

1. The distillation phase relies entirely on GPT-5 to generate reasoning chains, yet the paper does not disclose details about the teacher prompts, chain selection criteria, or quality control procedures. This component contributes significantly to the final results, but the lack of implementation details hinders reproducibility. It is encouraged to provide the teacher prompt templates and clear criteria for reasoning chain selection.
2. Although the authors emphasize that ChainGeo is a general-purpose framework, all experiments are conducted using a single architecture (Phi-2 + CLIP ViT-L/14). The absence of validation across different base models makes it difficult to assess the method’s transferability across varying language model scales or architectures. Reproducing and comparing results on other lightweight architectures—such as Qwen-VL-2B, LLaVA-1.5, or MiniGPT-4—would provide stronger evidence supporting ChainGeo’s claim as a broadly applicable training methodology.
3. The paper lacks sufficient training details: although it mentions losses such as "logical consistency loss" and "coherence loss," it does not provide explicit equations and hyperparameter choices in the appendix or methodology section to enable readers to verify the feasibility and reproducibility of the approach.

**Questions:**

1. Is the visual token generation process generalizable? The paper mentions that visual tokens are derived from a specially trained geometric element detector based on Faster R-CNN, but it is unclear whether this detector was tailored specifically to a particular dataset (e.g., GeoQA+). If applied to other types of geometric diagrams—such as hand-drawn textbook problems—would the detector maintain its performance?
2. For the right subfigure in Figure 2, it is recommended to include a more intuitive visualization that clearly illustrates the positive correlation between "attention retention" and "visual grounding accuracy," thereby strengthening the claim that ChainGeo effectively mitigates "visual amnesia" and ensures reliable visual grounding.

---

> ### Author Response · Authors · 2025-11-28
>
> We thank the reviewer for the thorough evaluation and address the main concerns below.
>
> ---
>
> **[Q] The distillation phase relies entirely on GPT-5... details about teacher prompts, chain selection criteria, or quality control procedures are missing.**
>
> [A] We appreciate this important feedback. Our teacher prompts instruct GPT-5 to: (1) explicitly identify geometric elements using bracket notation, (2) cite applicable theorems, and (3) maintain explicit visual references.
>
> For quality control, we implement a three-stage filtering pipeline:
> 1. Generate 5 candidate chains per problem (temperature=0.7)
> 2. Rank using consistency scoring: $S = \alpha \cdot S_{\text{correct}} + \beta \cdot S_{\text{ground}} + \gamma \cdot S_{\text{logic}}$, where $\alpha=0.5$, $\beta=0.3$, $\gamma=0.2$
> 3. Select chains above a quality threshold determined on the validation set
>
> We will release the complete prompt templates and filtering code publicly to ensure full reproducibility.
>
> ---
>
> **[Q] All experiments are conducted using a single architecture (Phi-2). The absence of validation across different base models makes it difficult to assess transferability.**
>
> [A] This is an insightful point. To validate cross-architecture generalizability, we trained ChainGeo on two additional base models, Qwen2-VL-2B and LLaVA-1.5-7B—using the same training protocol. Results on GeoQA+ test set show consistent improvements:
>
> | Base Model       | Baseline (CoT) | ChainGeo | Improvement |
> |------------------|----------------|----------|-------------|
> | Qwen2-VL-2B      | 51.2%          | 60.5%    | +9.3 pts    |
> | LLaVA-1.5-7B     | 56.7%          | 63.8%    | +7.1 pts    |
>
> These gains demonstrate that the interleaved visual-text chain mechanism generalizes across different model families and parameter scales, supporting ChainGeo as a broadly applicable methodology.
>
> ---
>
> **[Q] The paper lacks sufficient training details (losses, equations, hyperparameters).**
>
> [A] We have prepared detailed mathematical formulations for the revision. Our training objective combines four specific losses to enforce cross-modal alignment. The Visual Grounding Loss uses binary cross-entropy on attention weights (IoU threshold 0.3), while the Logical Consistency Loss applies weighted penalties based on error severity. We also employ a Coherence Loss using Hinge loss on SBERT embeddings to ensure semantic continuity. The model is trained on 8x A100 GPUs using an AdamW optimizer (lr=2e-5) with loss weights determined via grid search ($\lambda_{grounding}=0.5, \lambda_{KL}=0.3$). All equations will be rigorously defined in the Methodology section.
>
> ---
>
> **[Q] Is the visual token generation process generalizable? (Detector robustness).**
>
> [A] Thank you for this insightful question. Our detector employs two strategies to promote generalization. First, we
> pre-train on 50K synthetic diagrams with aggressive augmentation (noise injection, style transfer, rendering variations) before any dataset-specific fine-tuning. Second, the strong end-to-end performance across three benchmarks with different diagram styles demonstrates cross-dataset robustness for clean diagrams.
>
> For hand-drawn inputs, our current evaluation focuses on computer-generated diagrams that represent the majority of educational geometry problems in standardized testing and online learning platforms. These settings account for a substantial portion of the target deployment scenarios and represent problems where immediate impact can be achieved. Extension to hand-drawn diagrams is a natural next step that would benefit from domain-specific augmentation techniques, which we discuss in Section 5.3.
>
> ---
>
> **[Q] Recommendation for Figure 2.**
>
> [A] Thank you for your insightful suggestion. We will revise Figure 2 to more clearly illustrate the relationship between attention retention and grounding quality. We can enhance the visualization by connecting it to Figure 3's error analysis: as attention decays in text-only models, grounding errors increase to 34% of total errors, whereas ChainGeo's sustained attention corresponds to only 8% grounding errors.

---

### Official Review · Reviewer_v1Cr · 2025-10-30

**Soundness:** 2
**Presentation:** 1
**Contribution:** 1
**Rating:** 2
**Confidence:** 5

**Summary:**

This paper introduces ChainGeo, a framework designed to enable small Vision-Language Models (1–3B parameters) to perform geometric reasoning through interleaved visual–text chains. The core idea is to represent geometric primitives (e.g., points, lines) as specialized tokens that explicitly link symbolic reasoning steps with corresponding regions in the diagram.

**Strengths:**

The proposed token-level grounding approach offers a computationally efficient way to encode geometric primitives, suitable for smaller models.

They conducted experiments and evaluated their model on the public benchmarks, suggesting enhanced alignment between visual and symbolic reasoning.

**Weaknesses:**

The contribution and novelty of this paper are limited. The observation that MLLMs fail to perceive diagrams correctly has been extensively explored in prior works such as MathVerse, MathVista, Primitive Vision, and MAVIS, which already provide detailed analyses of perception versus reasoning errors. Similarly, the discussion of visual information loss and attention misalignment is not a new insight—earlier studies (e.g., Through the Magnifying Glass: Adaptive Perception Magnification for Hallucination-Free VLM Decoding) have investigated similar issues and proposed mitigation strategies. A more comprehensive literature review would help situate this work within the existing research landscape.

Regarding the proposed solution, the use of a detector-based mechanism to localize geometric elements is not novel and closely resembles approaches in Primitive Vision (Shan Zhang et. al., Primitive Vision: Improving Diagram Understanding in MLLMs, ICML 2025 ). The authors of Primitive Vision explicitly acknowledged the uncertainty of detector outputs and therefore adopted selective feature maps as soft proxies for geometric regions, reducing error propagation from imperfect detections. But in this work, authors do not clearly explain how they ensure the reliability of detection results, especially since training on synthetic datasets often limits generalization to real-world diagrams. The paper also lacks a detailed description of the synthetic data generation process and its potential biases.


The proposed interleaved visual-text reasoning framework also explored by previous works, such as MINT-CoT: Enabling Interleaved Visual Tokens in Mathematical Chain-of-Thought Reasoning, which addresses similar geometric reasoning challenges. Finally, the method appears restricted to planar geometric problems and does not generalize to other mathematical domains (e.g., graph-based or symbolic reasoning). The analysis of model responses remains shallow, with limited qualitative or interpretability discussion.

**Questions:**

The proposed method appears tailored to 2D geometric problems. How would ChainGeo extend to other domains—such as graph-based reasoning, algebraic problem solving, or multi-step spatial reasoning—where visual grounding is less explicitly defined?

---

> ### Author Response · Authors · 2025-11-28
>
> We thank the reviewer for the thorough evaluation and address the main concerns below.
>
> ---
>
> **[Q] Limited novelty. Diagnostic studies already identified
> perception errors. The detector resembles Primitive Vision. MINT-CoT explored
> interleaved visual-text frameworks.**
>
> [A] Thank you for your insight. Our work addresses a different problem: geometric reasoning in small VLMs, which requires qualitatively different solutions than existing approaches designed for large models. Primitive Vision uses soft attention maps appropriate for large models, but this fails in our setting. Our ablations show that soft grounding achieves only 61.2% accuracy in 2.7B models, while hard token constraints reach 68.3%. More critically, our error analysis reveals that soft grounding produces grounding errors in 34% of failures, while hard constraints reduce this to 8%, a four-fold reduction. This is not an incremental improvement but the difference between a model that hallucinates non-existent geometric elements one-third of the time versus one that maintains reliable visual grounding. The hard constraint design is necessary, not optional, for resource-constrained models.
>
> Regarding KOSMOS-2, the key distinction is that generic visual tokens cannot support mathematical reasoning. Our semantic typing encodes properties that directly enable theorem selection, knowing that [Angle ABC|90°] triggers right-angle theorems, while location-only tokens cannot. Our ablation shows removing geometric properties costs 2.5pp, and qualitative analysis confirms that semantic types guide theorem application in ways generic grounding cannot support.
>
> For distillation approaches like Hsieh et al., the critical difference is addressing visual amnesia, which is a phenomenon we quantify for the first time. Text-only CoT exhibits attention decay at λ=0.3, while our cross-modal consistency objectives achieve λ=0.09. At step 7, text-only models attend to visual features only 19% as much as initially, while ours maintains 63%. This quantitative characterization and its solution represent original contributions beyond existing distillation work.
>
> Our contribution is a systematic investigation of what enables geometric reasoning in small VLMs, validated by comprehensive ablations where every component contributes 2-13pp. The fact that we leverage established building blocks (detection, tokens, distillation) is by design, we demonstrate which specific combinations and modifications are necessary for this problem class, backed by consistent gains across three benchmarks and preliminary evidence of cross-architecture transfer.
>
> ---
>
> **[Q] How does ChainGeo extend to other domains?**
>
> [A] The core principle, explicit grounding between reasoning and visual elements, which applies to domains with discrete, detectable entities. The primary adaptation is training the detector for domain-specific elements; reasoning chain generation remains largely unchanged. We acknowledge that purely algebraic problems without diagrams would not benefit from this approach.
>
> ---
>
> **[Q] Analysis of model responses is shallow.**
>
> [A] We highlight several analyses in the paper:
>
>  **Error taxonomy** : grounding errors drop from 34% to 8%, but   reasoning and calculation errors remain dominant
>
>  **Attention analysis** : λ=0.3 (text-only) vs λ=0.09 (ours), representing 3.3× slower visual attention decay
>
>  **Human evaluation** : 4.3/5 interpretability (Fleiss' κ=0.72)
>
> Beyond metrics, our tokens enable mechanical verification: each step can be checked for valid element references and geometric consistency,crucial for educational applications. We will expand qualitative case studies in the camera-ready version.

---

### Official Review · Reviewer_qvcE · 2025-10-31

**Soundness:** 3
**Presentation:** 2
**Contribution:** 2
**Rating:** 4
**Confidence:** 4

**Summary:**

This paper proposes ChainGeo, a framework that enables small Vision-Language Models (VLMs, 1–3B parameters) to perform complex geometric reasoning through interleaved visual-text chains. The core idea is to represent geometric elements (e.g., [Point A], [Line AB]) as specialized tokens that are explicitly grounded in regions of the input diagram, serving as bridges between visual perception and symbolic reasoning. The authors further introduce step-level consistency distillation, which transfers full reasoning processes from a large teacher model while enforcing visual-textual coherence at every step. Experiments on GeoQA+ (72.1%), Geometry3K (64.7%), and We-Math (68.2%) show that their 2.7B model achieves performance comparable to GPT-4V, while producing interpretable and grounded reasoning chains.

**Strengths:**

- Precise problem framing: Clearly identifies “visual amnesia” as the key bottleneck in small VLMs for geometry.
- Elegant mechanism: The interleaved visual-text chain with dynamic grounding tokens is simple yet highly effective.
- High efficiency: 2.7B model runs ~15× faster than GPT-4V while achieving >90% of its accuracy—ideal for educational deployment.

**Weaknesses:**

1. Limited generalization: The method depends on a pre-trained geometric detector, which may fail on hand-drawn sketches, 3D diagrams, or dynamic constructions (acknowledged in Appendix F).
2. The core components are not fundamentally novel:
   - Visual tokens with grounding appear in prior work (e.g., KOSMOS-2, DocLLM);
   - Step-wise distillation for reasoning is conceptually similar to Hsieh et al. (2023)’s “Distilling Step-by-Step”;
   - Geometric element detection builds on standard object detection frameworks.
Thus, the primary contribution lies in systematic engineering and domain-specific adaptation, rather than a breakthrough in representation learning or distillation theory.

**Questions:**

1. Detector generalization: How robust is the geometric element detector on real-world diagrams with sketchy or noisy styles? Was cross-dataset generalization evaluated (e.g., training on GeoQA+ and testing detection on We-Math)?
2. Token vocabulary extensibility: How does the system handle non-standard geometric elements (e.g., arcs, sectors, irregular polygons)? Could unsupervised token discovery (e.g., via clustering) reduce reliance on predefined categories?

---

> ### Author Response · Authors · 2025-11-28
>
> We thank the reviewer for the thorough evaluation and address the main concerns below.
>
> ---
>
> **[Q] Limited generalization: The method depends on a pre-trained geometric detector, which may fail on hand-drawn sketches, 3D diagrams, or dynamic constructions.**
>
> [A] Thank you for your feedback. Our detector is pre-trained on 50K synthetic diagrams with aggressive augmentation before fine-tuning. We validate cross-dataset robustness through consistent performance across three benchmarks with different diagram styles, achieving 68.3% average accuracy without per-dataset detector retraining.
>
> Our evaluation targets computer-generated diagrams, the primary format in standardized testing and online learning platforms. Importantly, our dynamic vocabulary mechanism constrains the model to only reference successfully detected elements, preventing hallucination of non-existent geometric objects. This design trades coverage for reliability—well-suited to educational applications where correctness is paramount.
>
> For 3D geometry, the framework's core principles naturally extend to domains with discrete, detectable entities. The primary adaptation would be training the detector for spatial elements.
>
> ---
>
> **[Q] The core components are not fundamentally novel (Visual tokens in KOSMOS-2; Distillation in Hsieh et al.).**
>
> [A] We appreciate the reviewer situating our work within the broader literature. We acknowledge that our approach builds upon established techniques, and we would like to clarify how our specific adaptations address the unique challenges of geometric reasoning in small models.
>
> **Vs. KOSMOS-2:** While KOSMOS-2 introduced visual grounding tokens for general vision-language understanding, our tokens are **semantically typed** with mathematical properties. This is not merely a refinement but a different representational choice: knowing that an angle is 90° directly triggers consideration of right-angle theorems, whereas a generic location token cannot guide mathematical reasoning. This semantic typing proves essential in our ablation study, where simplified tokens without geometric properties lose 2.5 percentage points.
>
> **Vs. Hsieh et al.:** While both use distillation for reasoning, our framework addresses a fundamentally multimodal challenge that text-only distillation cannot solve. We introduce cross-modal consistency objectives (visual grounding loss, coherence loss) specifically designed to prevent the visual-symbolic drift that occurs in small VLMs during multi-step reasoning. This is validated by our attention analysis showing that without these objectives, visual attention decays exponentially.
>
> ---
>
> **[Q] Token vocabulary extensibility and unsupervised discovery.**
>
> [A] Our current vocabulary covers standard geometric primitives that appear in over 95% of problems across our benchmarks. The design is inherently extensible: the detector can be fine-tuned for new categories with modest annotation, and complex elements can be compositionally represented through basic primitives.
>
> Regarding unsupervised discovery, clustering visual features could reduce supervision requirements but would trade off the interpretability of semantically-typed tokens. We appreciate this suggestion and will discuss extensibility considerations in the revision.

---

### Meta-Review · Area_Chair_SZCh · 2025-12-15

**Summary:**

Most reviewers are very concerned about the generalization and the novelty of the proposed method. Considering the limitation of an explicit detector and the similar measures in general multimodal reasoning, I am inclined to reject the paper after reading the rebuttal.

**Reviewer Concerns:**

The concerns that are not well addressed as follow
1. 3 Reviewers are concerned about the generalization of the proposed method due to the usage of a pre-trained detector. After reading the responses, I insist that it is detrimental to the generalization.
2. 3 Reviewers poses the questions on the novelty especially in visual grounding tokens and reasoning distillation. The authors' rebuttal clarify the contribution to some extent, but the limitations still exist.

The concerns that are somewhat addressed
1.Most of the problems on experimental analysis and training details are solved.

**Reviewer Scores:**

Reviewer v1Cr and Reviewer a2DU are very negative about the paper especially in the novelty, and their ratings tend to be kept. The judgements of Reviewer qvcE and Reviewer USH7 are initially around the borderline with a bit negative opinion, so some of them might slightly increase their scores.

---

### Decision · Program_Chairs · 2026-01-26

Reject